# A genome-scale metabolic model of *Saccharomyces cerevisiae* that integrates expression constraints and reaction thermodynamics

Omid Oftadeh [1], Pierre Salvy [1,2], Maria Masid [1], Maxime Curvat[1,3], Ljubisa Miskovic[1] & Vassily Hatzimanikatis [1✉]

Eukaryotic organisms play an important role in industrial biotechnology, from the production of fuels and commodity chemicals to therapeutic proteins. To optimize these industrial systems, a mathematical approach can be used to integrate the description of multiple biological networks into a single model for cell analysis and engineering. One of the most accurate models of biological systems include Expression and Thermodynamics FLux (ETFL), which efficiently integrates RNA and protein synthesis with traditional genome-scale metabolic models. However, ETFL is so far only applicable for *E. coli*. To adapt this model for *Saccharomyces cerevisiae*, we developed yETFL, in which we augmented the original formulation with additional considerations for biomass composition, the compartmentalized cellular expression system, and the energetic costs of biological processes. We demonstrated the ability of yETFL to predict maximum growth rate, essential genes, and the phenotype of overflow metabolism. We envision that the presented formulation can be extended to a wide range of eukaryotic organisms to the benefit of academic and industrial research.

[1] Laboratory of Computational Systems Biotechnology, École Polytechnique Fédérale de Lausanne (EPFL), Lausanne, Switzerland. [2]Present address: Cambrium GmbH, Berlin, Germany. [3]Present address: Quotient Suisse SA, Eysins, Switzerland. ✉email: vassily.hatzimanikatis@epfl.ch

Eukaryotic organisms are extremely important in industrial biotechnology (e.g., *Saccharomyces cerevisiae*[1] and *Yarrowia lypolytica*[2]) and are host organisms for the production of fuels and specialty and commodity chemicals. Also eukaryotic, mammalian systems such as Chinese hamster ovary cells are the main platform organism used for therapeutic protein production[3]. In contrast to bacterial cells, the eukaryotes have compartmentalized cell structure to localize macromolecules with different biological tasks. This fundamental difference renders the engineering of the eukaryotes more complex and challenging. To help optimize and plan for industrial applications, complex biological systems such as these can be represented in silico by specific networks designed to capture key processes.

Metabolic networks are the most widely studied and modeled type of biological networks, with over 6000 genome-scale metabolic models (GEMs) reconstructed for archaea, bacteria, and eukaryotes[4,5]. One approach for analyzing these models is flux balance analysis (FBA), which is a constraint-based optimization technique, where the metabolic flux of individual reactions is computed in a metabolic network by formulating a linear optimization problem[6]. However, FBA can predict biologically irrelevant solutions, including cycles with unrealistically high fluxes[7] or thermodynamically infeasible solutions[8,9]. Despite its wide applicability, FBA cannot predict some important features of metabolic networks, such as those that account for limited catalytic capacity of enzymes or limitations in cellular expression systems.

To overcome some of the issues with FBA and eliminate unrealistic solutions, additional constraints that represent empirical or mechanistic evidence have been introduced. For example, thermodynamic-based flux balance analysis (TFA)[8,9] enforces the coupling between the directionality of each reaction with its corresponding Gibbs free energy to eliminate thermodynamically infeasible predictions. More importantly, TFA also directly integrates variables for the concentrations of metabolites, which enables the integration of metabolomics data. Genome-scale models with Enzymatic Constraints using Kinetic and Omics data (GECKO) are another FBA-based method that accounts for the limited catalytic activity of enzymes by inclusion of enzyme concentrations as variables[10]. Previous studies have shown that GECKO can capture a realistic maximum specific growth rate and the occurrence of overflow metabolism in *Saccharomyces cerevisiae*[10]. However, GECKO does not explicitly consider the cost of protein synthesis. Instead, it assumes that the fractions of peptides within a protein pool are inversely proportional to their molecular weight. The molecular weight represents the cost of the enzyme within the context of proteome allocation. However, the actual cost of enzyme synthesis is absent from the formulation. Therefore, GECKO fails to account for the competition for amino acids required for enzyme synthesis, which is an important part of the expression system.

Metabolic and Expression Models (ME-models) are another class of constraint-based models that include the cellular expression system in addition to metabolic and catalytic constraints[11–13]. ME-models include individual mRNA and enzyme concentrations as well as their cost of synthesis and cellular expression capacity. A new approach to construct ME-models, called Expression and Thermodynamics-enabled Flux (ETFL)[13], was recently proposed to address the significant drawback of needing to solve the nonlinear programming (NLP) problem. The approach approximates bilinear terms with a zeroth-order piecewise-linear function by discretizing growth and solving locally linearized mixed-integer problems instead of an NLP problem. Similar to published ME-models[11,14], the first ETFL model was developed for *Escherichia coli*. However, the ETFL formulation can readily be extended to the study of eukaryotic organisms.

*S. cerevisiae* is an industrially relevant organism[1,15] that is widely used for biological and medical research studies[16]. Several GEMs of this organism have been published over the years due to its ubiquity in metabolic engineering[17–22]. However, likely due to additional requisite considerations in modeling the compartmentalized cellular expression systems of eukaryotes, no ME-model of *S. cerevisiae* has been developed. The previous ME-models were constructed for bacteria[11–13], with one ribosome and one RNA polymerase being sufficient to represent the cellular expression machinery. In contrast, *S. cerevisiae* as a eukaryotic organism additionally has mitochondrial ribosomes and RNA polymerases. In this work, we extended the ETFL formulation and code for applicability to eukaryotic systems. In this new formulation, we account for the additional ribosomes and RNA polymerases within the eukaryotic mitochondrial expression system. We also included an allocation constraint for the fraction of proteins that are allocated to metabolism and cellular expression. Herein, we propose an ETFL model for *S. cerevisiae*, named yETFL, which is based on the extended ETFL formulation. The methodological advancements in ETFL provide avenues toward development of such models for the study of other eukaryotes.

## Results

**ETFL model of *S. cerevisiae*.** We present here yETFL, a ME-model of *S. cerevisiae* based on the ETFL formulation (Table 1). yETFL is constructed using the latest *S. cerevisiae* genome-scale model Yeast8[22]. Toward the generation of yETFL, we first performed a thermodynamic curation of Yeast8, which contains 1326 unique metabolites (a total of 2691 compartmentalized metabolites), 3991 reactions, 1149 genes, and 14 compartments (including the extracellular space). There are 2614 reactions associated to genes.

Information about the thermodynamic properties of reactions allows us to (i) integrate the available metabolomics and fluxomics data into the models, (ii) compute thermodynamically

**Table 1 Properties of the yETFL (variable biomass composition with thermodynamics) model created from Yeast8.3.4.**

| | |
|---|---|
| Growth upper bound ($\bar{u}$) | 0.75 h$^{-1}$ |
| Number of bins (N) | 128 |
| Resolution ($\bar{u}/N$) | 0.0058 h$^{-1}$ |
| Number of species | |
| - Metabolites | 2689 |
| - mRNAs | 1393 |
| - Peptides | 1393 |
| - rRNAs | 6 |
| Number of enzymes | |
| - Metabolic enzymes | 1059 |
| - RNA polymerases | 2 |
| - Ribosomes | 3 |
| Number of reactions | |
| - Metabolic | 2678 |
| - Transport | 1047 |
| - Exchange flux | 243 |
| - Transcription | 1393 |
| - Translation | 1393 |
| - Complexation | 1065 |
| - Degradation | 2458 |
| Thermodynamic data | |
| - Number of metabolites $\Delta G'^{\circ}_f$ | 1764 |
| - Number of reactions $\Delta G'^{\circ}_r$ | 1880 |

**Table 2 The nomenclature, number of variables, and constraints of different ETFL models.**

| Abbreviated name | Thermodynamics | Growth-dependent biomass composition | Number of variables | Number of constraints |
|---|---|---|---|---|
| EFL.cb | No | No | 43,527 | 70,918 |
| ETFL.cb | Yes | No | 66,714 | 92,338 |
| EFL.vb | No | Yes | 43,565 | 71,012 |
| ETFL.vb | Yes | Yes | 66,746 | 92,429 |

*EFL* Expression and Flux, *T* Thermodynamic, *cb* constant biomass composition, *vb* variable biomass composition.

consistent values of metabolic fluxes and metabolite concentrations, and (iii) determine thermodynamically feasible directionalities. Using the group-contribution method (GCM), we estimated the Gibbs free energies of formation for 1092 of 1326 total unique metabolites. We then estimated the Gibbs free energies for 1880 reactions in the Yeast8 GEM, which only includes reactions in an aqueous environment (see "Methods"). Yeast8 has 1304 reactions in the membrane compartments (nonaqueous environment). We did not apply thermodynamic constraints for these 1304 reactions as thermodynamic relations for membrane-associated metabolites require correction based on information about the nonaqueous environments, which is not always available.

In yETFL, we modeled the synthesis of 1059 enzymes coupled to 2588 of 2614 reactions with associated genes. The catalytic constraints are specified by coupling the reactions and the enzymes, which requires information on $k_{cat}$, or the enzyme turnover numbers. We found $k_{cat}$ values for 943 enzymes and approximated this number for a further 166 enzymes from the median $k_{cat}$ value in *S. cerevisiae* (see "Methods"). Of these enzymes, 77 were transporters associated to 167 transport reactions, there are 107 complexes among the enzymes, and the remainder are monomeric enzymes composed of a single peptide. A complexation reaction is considered for each enzyme to account for its synthesis from the constituent peptides. Operon structures were considered in the previous formulation of ME-models for the bacterial cells[14]. Similar to the original ETFL formulation, yETFL does not account for these structures since such mechanistic details were not necessary for the studies here. However, the details about the operon structures can be included in the ETFL formulation by expanding around the corresponding existing transcription steps, as it was done in the previous ME-models.

While one RNA polymerase and one ribosome can sufficiently represent bacterial expression system, in a eukaryotic cell such as *S. cerevisiae*, there are different RNA polymerases and ribosomes. Notably, the mitochondria have their own RNA polymerase and ribosome. The extended ETFL formulation, presented here, enables implementing multiple ribosomes and RNA polymerases, the latter of which includes: (i) the RNA polymerase II, which transcribes nuclear genes and (ii) the mitochondrial RNA polymerase, which transcribes the mitochondrial genes. The model also includes three ribosomes, where one ribosome is associated with mitochondrial genes and the other two ribosomes are associated with nuclear genes, but differ in their composition (see "Methods"). Altogether, yETFL includes 1149 metabolic genes from Yeast8 and an additional 244 genes that encode the composition of the aforementioned ribosomes and RNA polymerases.

To study the inclusion or exclusion of thermodynamic constraints and a variable or constant type of resource allocation ("Methods"), we developed four different types of models (Table 2). The inclusion of thermodynamic constraints is reflected by the presence of "T" in the name of the model (i.e., ETFL.cb and ETFL.vb), and the "cb" points to a version with a constant biomass composition, while "vb" indicates that the biomass composition is variable with growth. The number of variables and constraints in each model is detailed in Table 2. We used 128 bins to discretize the growth in the range of $[0, \mu_{max}]$, where $\mu_{max}$ is the maximum growth rate of *S. cerevisiae* as observed in rich growth medium as a conservative upper bound on growth rate. Here, we assumed that the highest growth rate *S. cerevisiae* can achieve in a normal condition is when it grows on the rich medium (see Salvy and Hatzimanikatis[13] for details). It is worth mentioning that $\mu_{max}$ can be increased or decreased by the users based on their needs to have higher resolution or higher range of the growth variation, respectively. Alternatively, we can increase the resolution by increasing the number of bins, but this entails increasing the number of variables and constraints. Using 128 bins to discretize the growth resulted in 135 (i.e., $128 + \log_2 128$) binary variables in the models without thermodynamic constraints, denoted as EFL.cb and EFL.vb. In the models with thermodynamic constraints, two binary variables were added per reaction to account for the directionality, which resulted in 8073 binary variables.

Similar models to yETFL developed for *S. cerevisiae* are GECKO models, ecYeast7[10] and ecYeast8[22], and WM_S288C[23], a whole-cell model. The Gecko models contain phenomenological constraints for proteome limitations. In contrast, yETFL is a fine-grained framework that accounts for proteome limitations mechanistically by integrating additional processes, such as transcription and translation. As a result, yETFL can predict parameters such as growth-condition-dependent biomass composition as well as transcription and translation machinery content, which cannot be done by the GECKO models. Moreover, the mechanistic representation of the expression system provides additional capabilities to yETFL to simulate the perturbations on the expression machinery, RNA transcripts, and gene copy numbers. Finally, in addition to proteomics data, which can also be integrated into the GECKO models, yETFL enables the integration of transcriptomics data.

The recently developed whole-cell model of *S. cerevisiae*, WM_S288C, decomposes cell functionality into 26 cellular processes[23]. yETFL includes three of those cellular processes, i.e., metabolism, RNA transcription, and protein translation, which makes WM_S288C broader in scope. A comparison of the common parts between both models shows that both approaches use a constraint-based framework to model metabolism, but they differ in the form they model expression. While WM_S288C uses ordinary differential equations, yETFL uses a constraint-based optimization framework. The simplified approach used in yETFL allows for an efficient analysis of cellular behavior for different physiological conditions and different strains overcoming the requirement of the vast number of biophysical parameters present in WM_S288C and which are highly dependent on the strain and the environmental conditions[24]. Furthermore, yETFL is able to simulate cellular processes under the macroscopic steady-state assumption and study the cell behavior in time intervals spanning a few hours[25], with a reasonable computational effort.

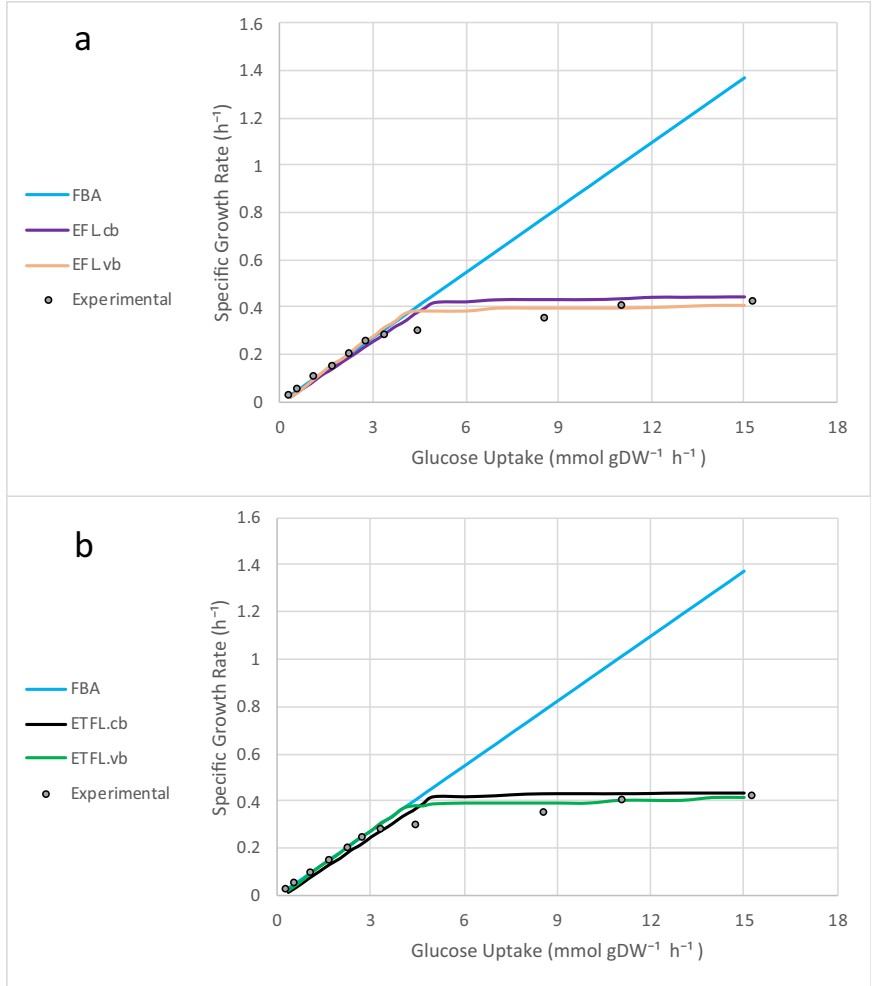

**Fig. 1 The maximum specific growth rate (h$^{-1}$) at different glucose uptake rates (mmol gDW$^{-1}$ h$^{-1}$). a** The models with thermodynamic constraints are compared against FBA. **b** The models without thermodynamic constraints are compared against FBA. The results are shown for ETFL models with constant (E[T]FL.cb) and variable (E[T]FL.vb) biomass composition. While using FBA, no maximum growth rate was observed, all ETFL models predicted a maximum for the growth rate, even in the presence of excessive substrate. The experimental data were taken from van Hoek et al.[27].

**Prediction of specific growth rate**. The cellular growth rate should plateau when high values of substrate uptake are attained, as limitations in the expression system and catalytic activity of enzymes cause shift the growth rate from a glucose-dependent limitation to an enzyme-dependent one. This phenomenon is described by established empirical models of microbial growth, where the growth shifts from nutrient limitation to proteome limitation[26]. However, standard FBA models predict that the growth rate increases linearly with increased carbon uptake. Since ETFL accounts for expression limitations, it is expected to predict this shift in the cellular growth rate.

We investigated the variations in growth rate with constant (E[T]FL.cb) and variable (E[T]FL.vb) biomass composition by examining the predicted maximum growth rate versus the glucose uptake (Fig. 1). With a constant biomass composition, the stoichiometric coefficients are constant in the growth reaction. Likewise, the stoichiometric coefficients change with growth in the variable composition. To account for this variation, the fractions and hence, stoichiometric coefficients of each biomass building block are determined based on experimental data. This way, we obtain a set of different biomass reactions, each associated with a specific growth rate. Then, we use a MILP optimization problem to determine which of the biomass reactions corresponds to the studied physiology (for more details,

see Salvy and Hatzimanikatis[13]). In both constant and variable biomass composition, and in contrast to FBA, the growth rate plateaued at higher values of glucose uptake rate, which is in accordance with the experimental results[27]. That is, we observed a shift from glucose-limited growth to proteome-limited growth. The maximum predicted growth rate was 0.44 and 0.41 h$^{-1}$ for E(T)FL.cb and E(T)FL.vb, respectively. Both agree with experimentally measured maximum growth rates reported in the literature, which are in the range of 0.4–0.45 h$^{-1}$ for different strains[28–30]. The accuracy of our predictions with experimental observations is important, as the maximum growth rate was highly overestimated in previously reported ME-models[12,13] that were developed for the other organisms, likely due to the lack of an allocation constraint on the total amount of metabolic enzymes (see Eq. (5)).

We observed small discrepancies in the maximal growth rate between the experimental data and the yETFL results for the glucose uptake rates, which ranged from ~4 to ~11 mmol gDW$^{-1}$ h$^{-1}$ (Fig. 1). One cause of these discrepancies might be the growth dependence of certain parameters, such as the ribosomal elongation rate. To avoid excessive constraints in the model and to preserve experimental observations in the feasible solution space, we used the highest reported values for ribosomal elongation rate, which typically corresponds to higher growth

rates[31,32]. Since our formulation accounts for growth-dependent parameters, we anticipate the facile integration of new information on the variation of the parameters with the growth rate into yETFL.

Another contributor to experimental and predicted discrepancies might be the regulation system that is used by *S. cerevisiae* during the transition from nutrient-limited to proteome-limited growth. Like in other ME-model formulations, we assume in yETFL that the cellular system evolved under selection pressure to maximize the growth rate. In this context, the regulatory network of *S. cerevisiae* can be seen as a control system that drives the metabolism toward optimality. Deviations from model optimality in transition regions are simply limitations of the regulatory system. Therefore, the predictive ability of the model can be enhanced by the addition of regulatory constraints from improved input on mechanisms and parameters that regulate the phenotypic transition.

**Gene essentiality analysis**. To investigate the quality of yETFL, we examined the ability of the model to predict which genes are essential for the cellular growth. We discovered that the gene essentiality results for metabolic genes were identical for the EFL.cb and FBA models (Table 3(a)). This includes 1149 genes associated with metabolic reactions in the Yeast8 model. We compared the predicted essentialities to the experimental observations, which were available for 5061 genes, to assess the quality of the model. However, these 5061 genes do not include all *S. cerevisiae* genes. The results in Table 3(a) show the essentiality of metabolic genes with the available experimental data. Compared to the FBA model, yETFL models have more genes that correspond to RNA polymerases and ribosomes (expression genes). We could not do gene essentiality for these 244 expression genes with FBA, as these genes are not associated to any function in the Yeast8 model. There are 222 expression genes with available experimental data that are represented alongside the metabolic genes in Table 3(b). We performed gene essentiality for 1393 genes in yETFL (compared to 1149 genes in Yeast8 that could be tested for gene essentiality), and we obtained a slight improvement in Matthews correlation coefficient (MCC) (Table 3). We also found that the integration of thermodynamic constraints into FBA or EFL.cb did not change the essentiality results.

**Crabtree effect**. Overflow metabolism is a shift from an optimal to a nonoptimal metabolic phenotype and is observed in different organisms at high growth rates[27,33,34]. Overflow metabolism in *S. cerevisiae*, also called the Crabtree effect, occurs when cells shift from pure respiration to a combination of respiration and fermentation in the presence of oxygen. This happens after cells reach a critical growth rate, which is strain-specific though can be estimated at about $0.3\,h^{-1}$. Because one hypothesis for why overflow metabolism occurs is proteome limitation[35,36] and because the yETFL model takes this into account, we therefore looked next at the ability of yETFL to predict this metabolic shift.

The Crabtree effect in *S. cerevisiae* cannot be predicted with FBA unless some ad hoc assumptions are made in the constraints or the objective function[36]. In contrast, we successfully predicted the shifts in fluxes at higher growth rates with yETFL, which considered limitations in the catalytic capacity of the enzymes and protein expression machinery (Fig. 2). In fact, yETFL could capture the shift in metabolism at high growth rates, where ethanol was secreted, and $CO_2$ production increased while $O_2$ consumption decreased. The model had good qualitative agreement with the experimental data acquired from aerobic, glucose-limited chemostat cultures[27].

The E[T]FL.vb models (see "Methods") presented an earlier onset of the Crabtree effect relative to the E[T]FL.cb models (Fig. 2). We can attribute the onset to the Yeast8 protein fraction used in E[T]FL.cb, which is close to the experimentally observed values at higher growth rates. Thus, the E[T]FL.cb models are less constrained than the E[T]FL.vb ones. In general, models with higher protein ratios are less tightly constrained. Hence, their maximum growth rate and the Crabtree effect occur at higher growth rates (Fig. 2). We also observed a slight deviation of the model predictions from the experimental observations in the transition region for the growth rates between 0.3 and $0.36\,h^{-1}$, the onset of Crabtree effect with the experimental data and yETFL, respectively (Fig. 2). A potential method to enhance the predictive ability of yETFL in light of these slight discrepancies would be through the inclusion of regulatory mechanisms by integration of regulatory constraints. Another next step would be to account for the growth dependence of more parameters. These improvements can be facilitated by further experimental investigations into *S. cerevisiae* physiology.

It is of note that yETFL was able to capture the Crabtree effect solely by integration of experimentally measured data and without ad hoc modifications in the model or the formulation. In an earlier study[10], an additional parameter was introduced to further constrain the availability of enzymes. Since the saturation rate of individual enzymes is not known, this parameter was introduced as the saturation rate of the total enzymatic pool and it was calculated by fitting the model predictions to the experimental data. Here, we captured the Crabtree effect without additional parameters, as yETFL explicitly accounts for the saturation rates of individual enzymes. Moreover, yETFL also allows for integration of experimentally observed saturation rates of individual enzymes by the addition of saturation parameters to the catalytic constraint of each enzyme. These parameters can then be found by fitting the model predictions to the experimental data, as has been reported[37]. Following a similar procedure, we can also integrate different experimental transcriptional and translational efficiencies into the model.

## Discussion

In this work, we developed a model for a eukaryotic organism, *S. cerevisiae*, by extension of the recently published formulation of ETFL to consider compartmentalized expression systems with separate ribosomes and RNA polymerases. This is the first model for yeast that includes RNA and enzyme concentration data, and this explicit simulation of expression broadens the applications of yETFL to the simulation of the impacts of different perturbations on cellular mechanisms. To test the accuracy of yETFL, we

**Table 3 Gene essentiality results for (a) only metabolic genes (FBA and E[T]FL.cb) and (b) metabolic and expression genes (E[T]FL.cb) compared with experimental results.**

| (a) | FBA,E[T]FL.cb (metabolic genes) | Predictions | |
|---|---|---|---|
| | MCC = 0.48 | Essential | Nonessential |
| Experimental | Essential | 53 | 106 |
| | Nonessential | 12 | 945 |
| (b) | E[T]FL.cb (metabolic and expression genes) | Predictions | |
| | MCC = 0.50 | Essential | Nonessential |
| Experimental | Essential | 72 | 118 |
| | Nonessential | 16 | 1132 |

Matthews correlation coefficient (MCC) was used as a metric to assess the quality of the predictions.

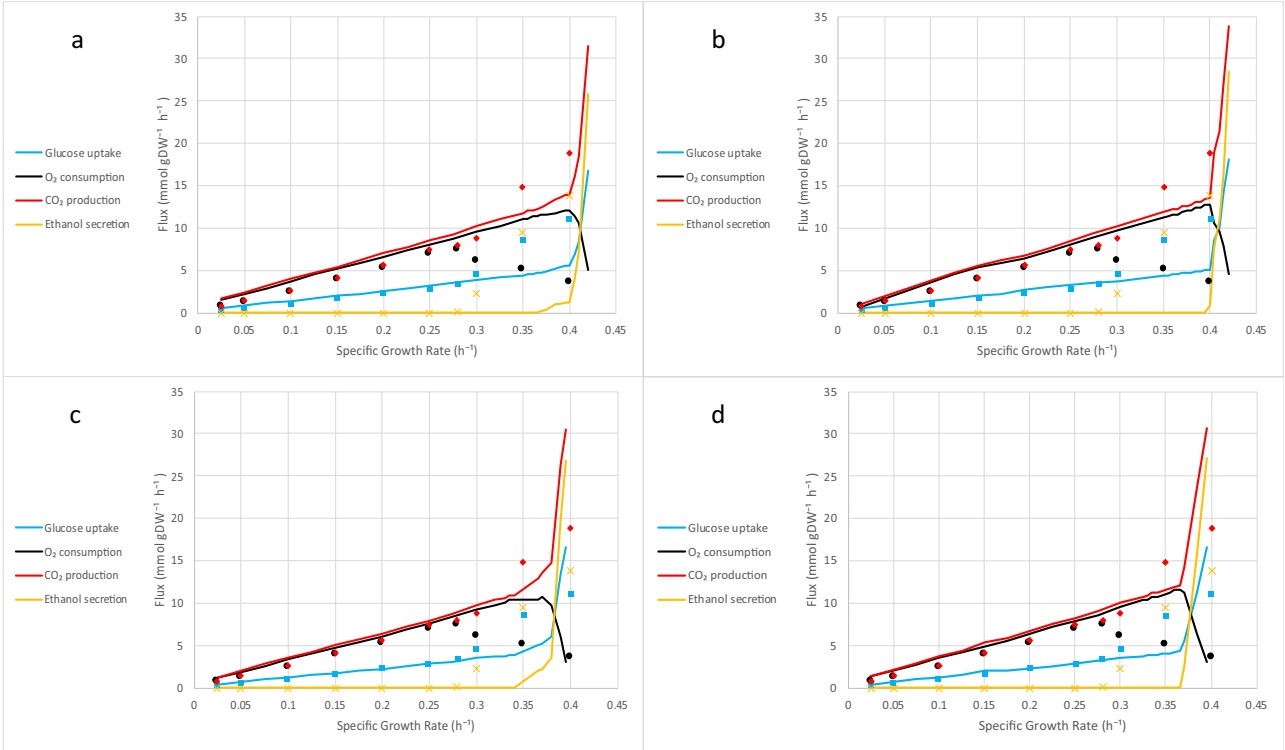

**Fig. 2 The simulation of the Crabtree effect. a** EFL.cb, **b** ETFL.cb, **c** EFL.vb, and **d** ETFL.vb models. In all cases, after a critical growth rate (0.35–0.39 h$^{-1}$ for different models), ethanol secretion was observed. Also, a shift in the fluxes of carbon dioxide production and oxygen consumption emerged, which indicated the shift of the organism from pure respiration to a combination of respiration and fermentation. The experimental data were taken from van Hoek et al.[27].

validated the predictions of the model against experimental data. Moreover, we reproduced the emergence of the Crabtree effect, and observed the secretion of ethanol in aerobic conditions without needing to integrate experimental data as with previous descriptions of the Crabtree effect[10].

Overall, a key advantage of the ETFL formulation is its direct extension to other types of analyses, such as the study of the Crabtree effect at the steady state as we have presented in this work. Future work in understanding the emergence of this effect in a dynamic setting, as previously shown for the *E. coli* overflow metabolism[25], will yield valuable insights on the optimality of the regulatory mechanisms in *S. cerevisiae*. We envision that this information can be applied to design industrially valuable strains. Also, yETFL can be used as a scaffold to integrate other biological networks, such as regulatory or signaling networks[5], as a vital step toward constructing a whole-cell model[38]. Finally, the extension of the ETFL formulation presented here is readily adaptable to any eukaryotic organism for which a well-curated GEM is available. The quality of the information about enzymes (i.e., catalytic rate constants and protein composition) will affect the quantitative predictions of the model, though new data are easily inputted into ETFL such that the predictions will always be as good as the available data. We envision that the availability of eukaryotic ME-models will improve the understanding and engineering of industrial hosts for the refinement and creation of better eukaryotic systems in biotechnology, for applications ranging from the production of fuels and commodity chemicals to therapeutic proteins.

## Methods

**Formulation of the ETFL model**. yETFL is based on the ETFL formulation, which was previously described in detail in Salvy and Hatzimanikatis[13] (for the full list of

the constraints in yETFL see Supplementary Tables S2 and S3). The ETFL constraints can be divided into five main categories:

- Metabolic constraints: enforce all metabolite and macromolecule concentrations to be at steady state. These constraints are the same as in FBA[6].
- Thermodynamic constraints: couple the directionality of reactions with their Gibbs free energy. These constraints are the same as in TFA[8,9].
- Catalytic constraints: define upper bounds on the reaction fluxes based on the enzymatic capacity of the associated enzymes. We account for the catalytic efficiency of each enzyme based on its turnover number ($k_{cat}$). A reaction might be catalyzed by different enzymes (i.e., isozymes) and the efficiency of these isozymes might be highly different. For example, if a reaction $i$ is catalyzed by two isozymes $j$ and $j'$, the corresponding catalytic constraint is:

$$v_i \le k_{cat,j}E_j + k_{cat,j'}E_{j'}$$

where $k_{cat,j}$ and $k_{cat,j'}$ reflect different catalytic efficiency of the isozymes. This way, the maximum catalytic capacity of the reaction is defined as the sum of the maximum catalytic capacities of individual isozymes. For the ribosomes and RNA polymerases, we model both free enzymes and enzyme complexes. For the metabolic enzymes, we do not account for enzyme mechanisms (e.g., Michaelis-Menten), and therefore making the distinction between free enzymes and enzyme complexes is not necessary. The enzyme mass balances constrain directly only the total amount of the enzyme, $E_{total}$. This quantity is used in the flux inequality constraints to express the maximal reaction velocity, $V_{max} = k_{cat}E_{total}$.

- Expression constraints: model the synthesis of mRNAs, peptides, and proteins, and constrain synthesis rates based on the limitations of transcription and translation machinery.
- Allocation constraints: determine the available amounts of DNA, RNA, and proteins in the cell. If experimental data are available, the ETFL formulation allows for modeling the growth-dependent abundance of these macromolecules during growth is not available, we assume that the ratio between these quantities is growth-independent, an assumption already made in FBA.

ETFL[13] is different from the other formulations of ME-models[12,14] in several aspects. On the one hand, only ETFL allows the integration of thermodynamic constraints and metabolomics data. Also, multiple expression systems were implemented only in ETFL. On the other hand, stable RNA splicing and operon

structure were considered in the previous formulations of ME-models, but not in ETFL (see Supplementary Notes for further details). The inclusion of ionic cofactors to form functional enzymes, i.e., metalloproteins, is partially considered in the previous formulations of ME-models, whereas ETFL lumps such requirements in the biomass reaction, alike the FBA models. Both formulations derived the mass balances for the macromolecules from the first principles, but rather in a different way. For example, in the other formulations of ME-models the enzymes are involved in the reactions as metabolites. In ETFL, however, the enzymes are coupled to the reactions based on their catalytic capacity (see Supplementary Notes and Supplementary Table S3). Finally, ETFL is a MILP formulation, which can be solved by conventional double-precision solvers, while the previous formulations of ME-models were nonlinear and only solvable with special quad-precision solvers.

### Data collection

*Genome-scale metabolic model (GEM)*. The most recent GEM of *Saccharomyces cerevisiae*, Yeast8[22], was used as a basis to construct the yETFL model. The latest published version of Yeast8 model, Yeast8.3.4, was obtained from the GitHub as it was provided by Laboratory of Systems and Synthetic Biology at Chalmers University (https://github.com/SysBioChalmers/yeast-GEM).

The following modifications to Yeast8 were made:

- Pseudometabolites defined for RNAs and proteins as well as pseudoreactions defined for their synthesis were replaced by the explicit expressions for RNA and protein synthesis (according to the procedure described in Salvy and Hatzimanikatis[13]).
- tRNAs and their reactions were adapted into a formulation that accounts for dilution effects, according to the ETFL procedure[13]. This is necessary as the dilution effect is not necessarily negligible for tRNAs.
- The biomass reaction was modified to account for growth-dependent composition, as discussed in detail in "Allocation data and constraints."

*Thermodynamic curation of Yeast8*. We used GCM[39] to determine the standard Gibbs free energy of formation in aqueous, ionic environments[40] for 1092 out of 1326 (82.4%) unique metabolites from Yeast8 (Fig. 3). We were not able to determine the thermodynamic properties for the remaining 234 metabolites because: (i) 89 metabolites (6.7%) represented abstract compounds, such as pools of proteins, nucleotides, lipid chains; (ii) 92 metabolites (6.9%) did not have a known molecular structure or they contained structural groups for which the estimated standard Gibbs energy of formation is unknown (e.g., acyl carrier protein group); and (iii) 53 metabolites (4%) contain groups with unknown energy in their composition. Using the standard Gibbs free energy of formation of compounds, we integrated the thermodynamic properties only for reactions in the aqueous solution. We estimated the standard Gibbs free energy of reactions for 1880 out of 2687 (70.0%) such reactions from Yeast8. The standard Gibbs free energy of reactions with at least one metabolite associated with a membranous compartment (including 1304 reactions) was not calculated using this procedure, as the standard Gibbs free energy of formation of compounds was determined for the aqueous environments (see Supplementary Notes).

*mRNA, peptide, and protein data*. The sequences for the peptides and mRNAs were obtained from the KEGG database[41]. Information about the stoichiometry of peptides forming enzymatic complexes in *S. cerevisiae* was obtained by combining available information in YeastCyc[42] and Complex Portal[43]. Turnover numbers ($k_{cat}$) were retrieved from BRENDA using functions provided by GECKO[10].

### Allocation data and constraints

We created yETFL models using either a constant or variable biomass composition. For constant biomass composition (E[T]FL.cb), we used the macromolecular fractions from the Yeast8 biomass reaction. The mass fractions for different macromolecules were calculated using the below equation:

$$f_k = \sum_{i \in M_k} \eta_i \mathrm{MW}_i. \tag{1}$$

For each type of macromolecule, $M_k$, $\eta_{i \in M_k}$ is the stoichiometric coefficient of the metabolites belonging to this macromolecule class in the biomass reaction, and $\mathrm{MW}_i$ is their molecular weight. For example, to find the protein fraction in the biomass, $f_{\mathrm{Prot}}$, the stoichiometric coefficients of individual amino acids were multiplied by their molecular weight to find their mass fractions in the biomass. The sum of these amino acid ratios indicates how much of the biomass is protein. By definition, the weight of biomass should be 1 g[44,45], i.e.,

$$\sum_{i \in \mathrm{BBBreactants}} \eta_i \mathrm{MW}_i + \sum_{j \in \mathrm{byproducts}} \eta_j \mathrm{MW}_j = 1. \tag{2}$$

In this equation, BBBreactants is the set of reactants in biomass reaction and byproducts is the set of all products except biomass.

When generating an ETFL model, it is important to remove protein and RNA metabolites from the biomass equation to prevent double counting of the metabolic requirements, since the explicit mRNA and peptide synthesis reactions already account for their respective participation in cell growth.

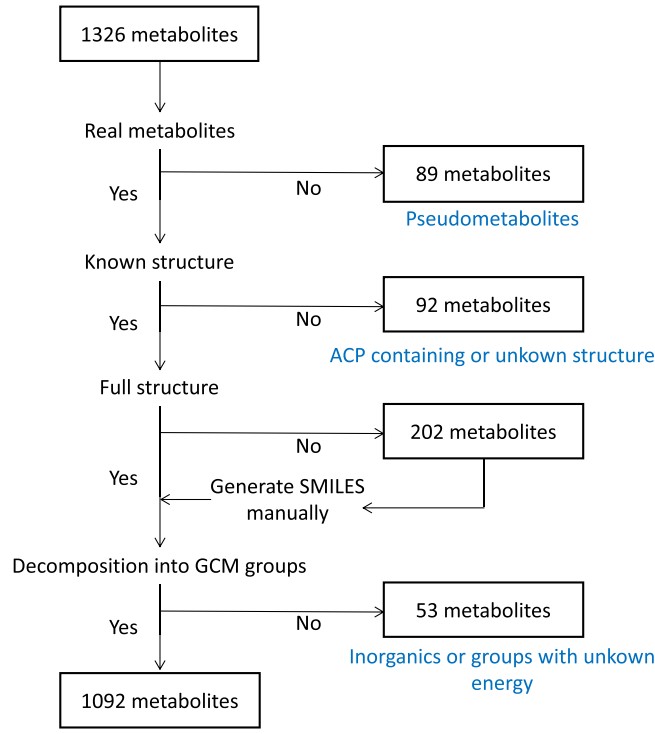

**Fig. 3 Schematic representation of the thermodynamic curation of the metabolites in Yeast8.** After excluding the pseudometabolites and ACP-containing proteins to which a concrete chemical structure cannot be assigned, the SMILES were generated or collected for the rest of the metabolites. Then, using the SMILES, each metabolite was decomposed into known functional groups. Finally, the standard free Gibbs energy of formation was estimated using the free Gibbs energy of formation of the groups that constitute each molecule. ACP acyl carrier protein, GCM group-contribution method, SMILES simplified molecular input line entry system.

In ETFL, we model the participation of macromolecules in the cellular biomass composition as follows:

$$\sum_j \mathrm{MW}_j E_j = P^m, \tag{3}$$

$$\sum_l \mathrm{MW}_l F_l = R^m, \tag{4}$$

where $P^m$ and $R^m$ are, respectively, the protein and RNA mass fractions in g gDW$^{-1}$, and $E_j$ and $F_l$ represent, respectively, the concentration of enzyme $j$ and RNA $l$ in mmol gDW$^{-1}$. $P^m$ and $R^m$ can either be constant (E[T]FL.cb) or variable and discretized (E[T]FL.vb). The constraints in Eqs. (3) and (4) ensure that Eq. (2) holds at different growth rates and different biomass compositions.

To create an E[T]FL.vb model, it is necessary to know the fraction of each biomass component at different growth rates. We gathered this information for *S. cerevisiae* by reviewing the literature (data available on the online yETFL repository, see "Data availability")[27,46,47]. Since the data are usually reported for a few particular growth rates, we resampled it using piecewise-linear interpolation.

*Protein allocation*. Since ME-models do not consider all the cellular tasks of proteins, ETFL defines a generic, so-called dummy protein to represent the fraction of the proteome not accounted for in the model[13], such as structural proteins, signaling proteins, or chaperones. However, since the dummy protein is not associated with a cellular function, the optimization procedure will apportion the whole protein content to the proteins that are associated with a cellular task (i.e., metabolic enzymes, ribosomal peptides, and RNA polymerase). Consequently, the concentration of the latter proteins is overestimated, which results in over-estimating the maximum growth rate, and the Crabtree effect emerges at higher growth rates. To realistically account for enzyme participation in the proteome, we can define $\varphi$, the proportion of proteins that is associated with a metabolic task, in the total protein content of the cell. Then, we can add the following constraint in

Flowchart text:
1326 metabolites
Real metabolites — Yes / No → 89 metabolites — Pseudometabolites
Known structure — Yes / No → 92 metabolites — ACP containing or unknown structure
Full structure — Yes / No → 202 metabolites
Generate SMILES manually
Decomposition into GCM groups — Yes / No → 53 metabolites — Inorganics or groups with unknown energy
1092 metabolites — Found formation Gibbs energies

the optimization problem

$$\sum_{j \neq \text{dummy protein}} \text{MW}_j E_j = \varphi \cdot P^m. \tag{5}$$

This way, the constraints in Eqs. (3) and (5) enforce the optimization procedure to allocate a fraction of the proteome, i.e., $(1 - \varphi)$, to the proteins with cellular functions not considered in the model, i.e., dummy protein. We used the latest protein abundance dataset for *S. cerevisiae* available in PaxDB[48] to compute this fraction as $\varphi = 0.55 \, \text{g} \, \text{g}_{\text{protein}}^{-1}$.

*DNA.* The growth dependence of the DNA abundance in the cell was modeled as proposed in the original ETFL formulation[13].

*Carbohydrates, lipids, and ions.* To consider the growth dependence of the biomass composition, we introduced the variation of the other biomass components in the ETFL formulation. The growth dependence of carbohydrates, lipids, and ions is implemented in a similar way to the one of DNAs, RNAs, and proteins in the original ETFL formulation. Specifically, we first discretize the growth into bins and then we use a MILP optimization to determine the discretized growth value (bin) that corresponds to the studied physiology. Then, we defined a metabolite pool for each of these macromolecules. In Yeast8, each biomass component is attached to a pooling reaction that transforms the sum of specific metabolites (e.g., all carbo-hydrate metabolites) into a single metabolite pool (e.g., carbohydrate). The mass balance equation for these modeling metabolites is the following:

$$\frac{d[X_i]}{dt} = \eta_i^{\text{biomass}} \mu + \eta_i^{\text{pool}} v_i^{\text{pool}}, i \in \{\text{Carbohydrate, Lipid, Ion}\}, \tag{6}$$

where $v_i^{\text{pool}}$ is the flux through the pooling reaction, and $\eta_i^{\text{pool}}$ and $\eta_i^{\text{biomass}}$ represent stoichiometric coefficients of the modeling metabolite $i$ in the pooling and biomass reactions, respectively. When it is desired to model a growth-dependent stoichio-metric coefficient in the biomass reaction, the said stoichiometric coefficient can be redefined as a function of $\mu$ and calculated as follows:

$$\eta_i^{\text{biomass}} = \eta_{i,\text{ref}}^{\text{biomass}} \frac{X_{u,i}^m}{X_{\text{ref},i}^m}, i \in \{\text{Carbohydrate, Lipid, Ion}\}. \tag{7}$$

In this equation, $X_{u,i}^m$ is the discretized mass fraction of component $i$ in the discretized growth bin $u$, following notations from Salvy and Hatzimanikatis[13]. $\eta_{i,\text{ref}}^{\text{growth}}$ is the stoichiometric coefficient in the biomass reaction, and $X_{\text{ref},i}^m$ is the mass ratio of component $i$ in a reference model (e.g., FBA).

**Ribosomes and RNA polymerases.** To model the ribosomes and the RNA polymerases, information about their constituting peptides, ribosomal RNA, and catalytic rate constants is required. To consider the eukaryotic complexity of *S. cerevisiae*, we defined multiple RNA polymerases and ribosomes in yETFL (Table 1)

- RNA polymerase: similar to the other eukaryotes, *S. cerevisiae* has three different types of nuclear RNA polymerases. However, most of the mRNA transcripts are transcribed by RNA polymerase II[49]. In yETFL, we implemented this nuclear RNA polymerase, and we modeled such that all the nuclear genes could be transcribed only by this enzyme, similar to the previous work[13]. For mitochondrial genes, we defined a mitochondrial RNA polymerase, which was characterized by its own composition and kinetic parameters[49].
- Ribosome: the structure of the cytosolic ribosomes in *S. cerevisiae* contains four ribosomal RNA (rRNA) molecules encoded by four different genes. In addition to these four rRNAs, the cytosolic ribosomes contain 78 peptides encoded by 137 genes[50]. Out of 78 peptides, 19 are encoded by a single gene and 59 peptides are encoded by either of two alternative genes. To account for alternative ribosomal peptides, we defined two sets of genes: set A containing 59 genes encoding for the 59 peptides (designated with "A" in their standard names, e.g., RPL1A), and set B containing the alternative genes of set A (designated with "B" in their standard names, e.g., RPL1B). Then, we constructed two cytosolic ribosomes, one where we constructed the 59 peptides using the set A and the other where we used the set B. We assumed a similar elongation rate for both cytosolic ribosomes. The two modeled ribosomes represent only two out of many possible combinations of the peptides from sets A and B. Implementing these $2^{59}$ possibilities is currently computationally intractable. Here, as a first approximation, we decided to keep the two ribosomal compositions because we wanted to be inclusive of all ribosomal genes while having a realistic production cost. The other possible combinations can be readily integrated into the model by adding similar constraints to what we have already included for the two modeled ribosomes. A mitochondrial ribosome was also defined to translate mitochondrial genes. This ribosome is composed of two rRNAs and 78 peptides[52].

Further details about the expansion of the ETFL formulation to implement multiple expression systems are provided in Supplementary Notes.

**Modifying the growth-associated maintenance (GAM).** The energetic cost of growth, including maintenance of the cell and polymerization of the macromolecules[53], is quantified in genome-scale models using the GAM. In ETFL, we consider the energetic cost of protein synthesis explicitly, and this cost should be removed from the GAM to avoid the overestimation of energetic requirements in the polymerization of peptides (Eq. (8)).

$$\sum_{aa_i \in A} \eta_{aa_i}^l \text{tRNA}_{aa_i}^{\text{charged}} + 2L_{aa}^l(\text{GTP} + \text{H}_2\text{O}) \rightarrow \text{Pep}_l + \sum_{aa_i \in A} \eta_{aa_i}^l \text{tRNA}_{aa_i}^{\text{uncharged}} + 2L_{aa}^l(\text{GDP} + \text{Pi} + \text{H}^+), \tag{8}$$

where $aa_i$ is the $i$th amino acid, $\eta_{aa_i}^l$ represents its count in the $l$th peptide ($\text{Pep}_l$), and $L_{aa}^l$ is the length of the peptide in amino acid.

Since 2 moles of GTP are needed to attach 1 mole of amino acid to the peptide (Eq. (8)), and from

$$\text{ATP} + \text{GDP} \rightarrow \text{ADP} + \text{GTP}, \tag{9}$$

1 mole of ATP is required to produce 1 mole of GTP. Therefore, we can deduce that peptide polymerization requires 2 moles of ATP per 1 mole of amino acid.

We also know that the stoichiometric coefficients of amino acids in the biomass reaction of Yeast8 give information on how many mmol gDW$^{-1}$ of each amino acid are required to produce 1 g of biomass. From there, it is straightforward to compute the total amount of amino acids (~4.1 mmol) required for the production of 1 g of biomass. Combined, we can calculate that to produce 1 g of biomass, the energetic cost is $2 \times 4.1 = 8.2$ mmol gDW$^{-1}$ of ATP for peptide synthesis, which we removed from the GAM.

**Gene–protein–reaction coupling.** Coupling the reactions in metabolic networks with their enzymes is the most important step in the process of creating an ETFL model. Ideally, assigning enzymes to reactions requires information about: (i) gene–protein–reaction rules, (ii) catalytic rate constants ($k_{\text{cat}}$), and (iii) type and stoichiometry of the peptide assembly into enzymes. Whenever we did not have access to all required information, we made the following assumptions (Fig. 4):

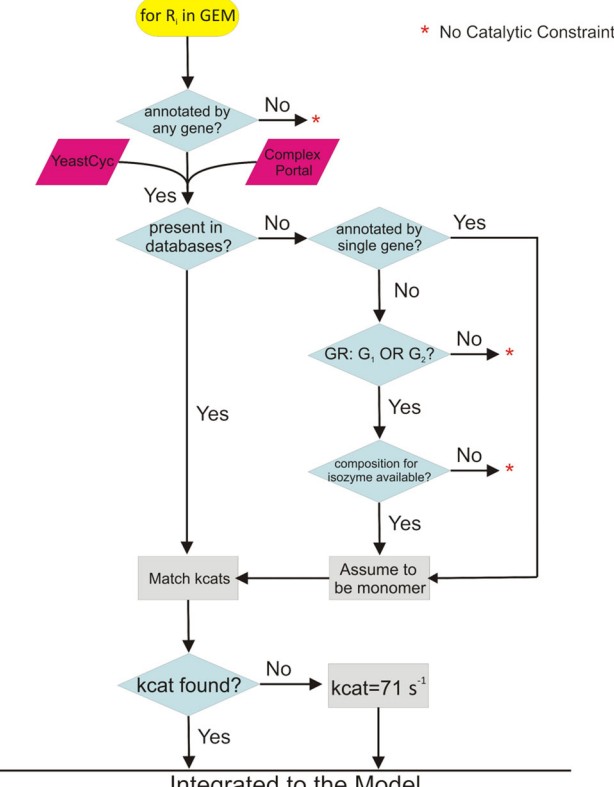

**Fig. 4 Workflow for the integration of enzymes into the model.** The enzyme composition for the complex enzymes was sourced from YeastCyc and Complex Portal. In cases that the enzyme composition was found for one isozyme but not for the other(s), we assumed a similar composition for all isozymes (e.g., all were assumed to be dimers, trimers, etc.). We used the function MatchKcats from GECKO[10] to find turnover numbers ($k_{\text{cat}}$). If the $k_{\text{cat}}$ cannot be found for an enzyme, we used the median of the $k_{\text{cat}}$s in *S. cerevisiae* as an approximation.

- We assumed similar composition for isoenzymes if composition information was only available for one of them. For example, if one of the isoenzymes is a dimer, the other is also assumed to be a dimer.
- We assumed that monomeric enzymes catalyze reactions (i) that depend on a single gene, and (ii) for which information about their enzyme composition was not available.
- If an enzyme peptide composition is identified, either from databases or by approximation, but its $k_{cat}$ was not found, we set the $k_{cat}$ equal to $70.9\,\mathrm{s}^{-1}$, which is the median for $k_{cat}$s in *S. cerevisiae*[10].
- While the reactions that transport a metabolite from one compartment to another one are associated with genes, their $k_{cat}$ information is scarce. As a result, these reactions were not catalytically constrained in similar models such as GECKO[10]. We set $k_{cat}$ of the proteins that catalyze these reactions to a large number ($1E + 9\,\mathrm{h}^{-1}$), which ensures that these reactions are not catalytically constrained and only the gene–protein–reaction relationship is preserved. We also checked the impact of constraining the transport reactions. To this end, these reactions were constrained by the median $k_{cat}$, but no significant change was observed in the results.

**Gene essentiality analysis**. We used gene essentiality analysis[54] to assess the quality of yETFL. The ETFL formulation enables single-gene knockouts by blocking the flux through transcription reaction for each gene. The predicted essential genes were compared against experimental data for *S. cerevisiae* obtained from http://www-sequence.stanford.edu/group/yeast_deletion_project/downloads.html. Before deleting the genes, the culture medium was modified according to Lu et al.[22]. Briefly, the minimal medium supplemented with amino acids and nucleotides was used for the simulations, and the model was allowed to uptake glucose as the sole carbon source. The MCC was used as a metric to evaluate the quality of predictions for FBA and ETFL because of its robustness to the imbalance in the number of essential and nonessential genes. MCC can take values from $-1$ to 1, where values of MCC close to $-1$ indicate predictions opposed to the ground truth, 0 random predictions, and 1 perfect predictions.

**Chemostat simulations**. The results of this paper were obtained by simulating the cell growth as a function of different carbon uptake rates. This allows the exhibition of proteome-limited behavior and overflow metabolism in the presence of excess glucose. For all simulations, the model was allowed to uptake glucose as a carbon source, some essential inorganic compounds, and oxygen. To prepare the model for the simulations, it was modified as described previously in Sánchez et al.[10].

To capture the Crabtree effect, the substrate uptake rate was minimized for different values of the growth rate. Then, we fixed the values of the substrate uptake rates at the computed minima and minimized the total fluxes[55] and then the total enzyme concentrations[10], consecutively, to account for the parsimonious enzyme usage. Finally, the Chebyshev center of the enzyme space was used as a representative solution[25].

**Reporting summary**. Further information on research design is available in the Nature Research Reporting Summary linked to this article.

## Data availability
The supporting data used in this study[51] are available in Zenodo (https://doi.org/10.5281/zenodo.4778047). This data is collected from public databases including KEGG (https://www.genome.jp/kegg), yeastCyc (https://yeast.biocyc.org), and Complex Portal (https://www.ebi.ac.uk/complexportal/home). The other parameters that are set inside the code are provided in Supplementary Tables S1 and S4–S7.

## Code availability
The code was implemented in Python 3.7, and the commercial solver Gurobi was used to solve the MILP problems. The code relies on the ETFL[13] and pyTFA[56] packages, which use COBRApy[57] and Optlang[58]. The code to generate yETFL models and reproduce the results of this paper is freely available at https://github.com/EPFL-LCSB/yetfl and https://gitlab.com/EPFL-LCSB/yetfl (the code is also deposited in Zenodo to provide a reference to the version used in this study[59]).

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

## Acknowledgements
The authors would like to thank Dr Kaycie Butler for her help in improving the wording and structure of this manuscript. This work has received funding from the European Union's Horizon 2020 research and innovation programme under Grant Agreement No. 814408 (O.O.), the Swiss National Science Foundation under Grant Agreement 200021_188623 (O.O.), the European Union's Horizon 2020 Research and Innovation Programme under the Marie Skłodowska-Curie Grant Agreement No. 722287 (P.S.), the European Union's Horizon 2020 research and innovation program under the Marie Skłodowska Curie Grant Agreement No. 675585 (M.M.), and the École Polytechnique Fédérale de Lausanne.

## Author contributions
O.O., P.S., and V.H. designed the study. O.O. and P.S. wrote the code to adapt ETFL to eukaryotic organisms. O.O. ran the simulations and did the enzymatic data curation. O.O., L.M., and V.H. analyzed the results and provided the discussion. M.C., M.M., and L.M. performed the thermodynamic curation of the Yeast8 GEM. O.O., P.S., M.M., L.M., and V.H. wrote and reviewed the manuscript.

## Competing interests
The authors declare no competing interests.
