## [Peer Review File · Nature Communications]

Reviewers' Comments:

Reviewer #1:

Remarks to the Author:

In this manuscript, Oftadeh et al present a metabolic-expression (ME) model for *Saccharomyces cerevisiae*. ME models are extensions of genome-scale metabolic (M) models, which in addition to metabolic reactions, account for gene expression processes related to metabolism and cellular growth. This work is an extension of a new ME model formulation referred to as Expression and Thermodynamic Flux (ETFL) for prokaryotes, which was recently published by the same group and applied to *E. coli*. In this study, the authors extend ETFL to eukaryotes and demonstrate its applicability by constructing a ME model for *S. cerevisiae*. This extension (named γ ETFL) accounts for additional ribosomes and RNA polymerases within mitochondria and for the fraction of proteins allocated to metabolism and growth. The authors verified the accuracy of this model by predicting the growth of *S. cerevisiae* for different glucose uptake rates and by performing gene essentiality analysis, and comparing with experimental data, and reported close agreements between the two. Additionally, the authors demonstrated that this model is able to capture the Crabtree effect without addition of any ad hoc constraints to the formulation.

The novel contribution of this paper is proposing the first ME model for a eukaryotic cell, which paves the way for constructing such models for other eukaryotes as well. Additionally, given the wide applicability of *S. cerevisiae* for both basic and applied science and for industrial applications, the constructed ME model in this study will be of interest to a wide range of audience. Overall, this is a well-written paper and a step in the right direction. Addressing the following concerns can improve the quality of the manuscript:

1. Given that a whole-cell model of *S. cerevisiae* already exists (Ye et al, *Biotechnol Bioeng*, 2020, PMID: 32022245), at least a descriptive comparison of the γ ETFL model and this existing model seems necessary.
2. Given that (γ)ETFL is a new ME model formulation, a head-to-head comparison of how different biological processes are formulated here compared to existing ME models (refs 11, 12 and 14 of the manuscript) could be very helpful. For example, it is not clear whether (γ)ETFL accounts for reactions related to transcription and translation initiation or operon structures ("transcription units") implemented in existing ME models. Additionally, references for synthesis reactions or expression constraints are missing in the original ETFL paper (ref 13) making it unclear as to whether these were taken from existing ME models (refs 11, 12, 14) or were proposed for the first time in that paper. The authors can provide a summary of the processes or features in existing ME models that are not accounted for in (γ)ETFL or vice versa. Additionally, for processes that are accounted for in both formulations, they can comment on how their treatment/formulation might differ between the two modeling paradigms.
3. Can the proposed ME model capture the relative catalytic efficiency of isozymes? While in metabolic networks only the presence or absence of the enzymes are accounted for, we know that, in practice, the catalytic efficiency of the isozymes could be very different, i.e., the reaction rate could be different if one isozyme is removed compared to the other. How about when both isozymes are present? Will they have an additive effect on the reaction rate in (γ)ETFL?
4. Line 74: ETFL does not "avoid" bilinear terms. It just approximates bilinear terms with a zeroth-order piecewise linear model.
5. Equation (6): Each modeling metabolite i (e.g., carbohydrate) is produced in the pooling reaction and is consumed in the biomass reaction. These are taken care of by sign of the stoichiometric coefficients (positive in the former and negative in the latter). Therefore, the minus sign in this equation (before the second term) should be changed to a plus sign.
6. Carbohydrates, lipids and ions (line 406): It is not clear how the growth-dependent biomass composition is implemented in the formulation. This sounds like an "if else" condition, e.g., if $0 \leq \mu \leq a$ then use these fractions else if $b < \mu \leq \mu_{\max}$ use these other fractions, which requires defining new binary variables. Is this how these are implemented this in (γ)ETFL?

7. Is the mass fractions constraint (Equation 2) for E(T)FL_cb added as a constraint to the formulation?
8. Looking at the original ETFL formulation (ref 13), I see that the authors include total capacity constraints for ribosome and RNA polymerase but I do not see any such constraints for metabolic enzymes (Enz_j). Is this a simple error or these constraints are not considered for metabolic enzymes for some reasons?
9. Table 3's caption: I think it is "Mathews" not "Mathew's" correlation coefficient.

Reviewer #2:

Remarks to the Author:

In their manuscript, Oftadeh et al. have translated their ETFL modeling approach to *S. cerevisiae*: yETFL. This is a significant advance, as it is the first ME model of *S. cerevisiae* (although see my later comment). The major value of this model would be reproducible reuse by other research groups, who might want to address a wide variety of research questions, but unfortunately for this purpose the model is inaccessible and significant detail on how the model was reconstructed is missing. In addition, the applications and validations shown are not particularly advanced, similar results have been shown with other formulations before and would rather be suited in Supplementary Material as necessary validation. The paper would improve significantly if the authors would be able to show one or several use cases where the model is providing new insight in yeast metabolism and its regulation.

1. Line 87: "no ME-model of *S. cerevisiae* has been developed", while this is true to my knowledge, there is a whole-cell model WM_S288C (doi:10.1002/bit.27298) which one could include an ME-model as part of it. While the scope and application is different, it is worthwhile to mention and to discuss it in relation to yETFL.
2. Line 104 (and 331): it is unclear what curations were applied to Yeast8. Thermodynamic properties were gathered and used in the TFA part, but what curations were done on the model itself? The GitHub repository contains yeast8_thermo_curated.mat, but the authors should provide a list of curations in supplementary material.
3. Line 137: it is unclear why only two possible compositions of ribosomes were modelled, are there not many different possible combinations of subunits?
4. Line 148: why is rich growth medium used to define μ_{\max} in the model, as further model simulations are performed with minimal medium? Does rich medium not imply that many enzymes will not be required to sustain growth, and therefore give an underestimation of enzyme demand when simulated with minimal medium?
5. Line 175: please clarify that "previously reported ME-models" refers to ME-models in general, not a non-existent *S. cerevisiae* ME-model.
6. Line 186: please document in Supplementary Material what parameter values were used. This also refers to any other parameter that was set in yETFL, the Materials and Methods section does not give sufficient information on this.
7. Line 194: verb missing.
8. Line 207: unclear what is meant with "However, this is not comprehensive of *S. cerevisiae* genes"

9. Line 226: swap to "looked next"

10. Line 265: in the introduction GECKO is mentioned as the closest alternative to yETFL. What is missing from the discussion is then how yETFL is better than GECKO? The results in the manuscript show that yETFL performs as well as GECKO (albeit without saturation factor), but what are things yETFL can do / predict / analyze that cannot be done by the GECKO model?

11. Line 293: the Materials and Methods section is not detailing all relevant information. It is probably most suitable to provide this in Supplementary Material instead. This includes what curations were performed on Yeast8 and what parameters were set, as both mentioned above. The data provided on Zenodo seems like a random selection from the whole GitHub repository (why not have the whole repository backed-up on Zenodo?). Why is the curated model distributed as MATLAB file, particularly considering pyTFA runs on python? The thermo_data_yeast8.mat file does not have sufficient meta data explaining its content. There should be a much more detailed explanation of data presentation and how this was used in establishing yETFL.

12. Line 526: having a separate branch, with even dev_ in the name, does not seem like a good solution.

13. GitHub: I was unable to run the code (Ubuntu 20.04 on WLS2, python 3.8.5). pipenv freeze output is included at the end of the review. When following the instructions in README_yETFL.txt, when sequentially running tutorials/helper_gen_models_yeast.py gives the following error message:

Traceback (most recent call last):

```
File "helper_gen_models_yeast.py", line 16, in <module>
from etfl.core import Enzyme, Ribosome, RNAPolymerase, ThermoMEModel, MEModel
File "/home/XXXX/yETFL/etfl-dev_yetfl/etfl/core/__init__.py", line 1, in <module>
from .enzyme import Enzyme, Ribosome, RNAPolymerase
File "/home/XXXX/yETFL/etfl-dev_yetfl/etfl/core/enzyme.py", line 19, in <module>
from Bio.Alphabet import DNAAlphabet, ProteinAlphabet
File "/home/XXXX/.local/share/virtualenvs/yETFL-gNQIO2nY/lib/python3.8/site-
packages/Bio/Alphabet/__init__.py", line 20, in <module>
raise ImportError(
ImportError: Bio.Alphabet has been removed from Biopython. In many cases, the alphabet can simply
be ignored and removed from scripts. In a few cases, you may need to specify the ``molecule_type``
as an annotation on a SeqRecord for your script to work correctly. Please see
https://biopython.org/wiki/Alphabet for more information.
```

Alternatively, a Docker container is provided, although wide-spread use of (y)ETFL is not helped if it only runs as Docker. Regardless, following the instructions (on the master branch of the repository) result in an error message when ". run" is run in etfl/docker:

```
ERROR: File "setup.py" not found. Directory cannot be installed in editable mode: /src/optlang
```

Therefore, I have not been able to run the code and have therefore not been able to evaluate its reproducibility.

```
#### pip freeze output:
appdirs==1.4.4
attrs==20.3.0
biopython==1.78
bokeh==2.3.0
certifi==2020.12.5
cobra==0.21.0
colorama==0.4.4
commonmark==0.9.1
decorator==4.4.2
depinfo==1.7.0
diskcache==5.2.1
# Editable install with no version control (ETFL==0.0.2)
-e /home/XXXX/yETFL/etfl-dev_yetfl
future==0.18.2
h11==0.12.0
httpcore==0.12.3
httpx==0.17.1
idna==3.1
importlib-resources==5.1.2
iniconfig==1.1.1
Jinja2==2.11.3
MarkupSafe==1.1.1
mpmath==1.2.1
networkx==2.5
numpy==1.20.1
optlang==1.4.4
packaging==20.9
pandas==1.2.3
Pillow==8.1.2
pluggy==0.13.1
py==1.10.0
pydantic==1.8.1
Pygments==2.8.1
pyparsing==2.4.7
pytest==6.2.2
pytfa==0.9.3
python-dateutil==2.8.1
python-libsmb==5.19.0
pytz==2021.1
PyYAML==5.4.1
rfc3986==1.4.0
rich==6.2.0
ruamel.yaml==0.16.13
ruamel.yaml.clib==0.2.2
scipy==1.6.1
six==1.15.0
sniffio==1.2.0
swiglpk==5.0.3
sympy==1.7.1
toml==0.10.2
tornado==6.1
tqdm==4.59.0
typing-extensions==3.7.4.3
```

xlrd==2.0.1

Reviewer #3:

Remarks to the Author:

The Authors have previously presented an approach to extend genome-scale metabolic models with expression and thermodynamic constraints for a prokaryotic model organism *E. coli* <https://www.nature.com/articles/s41467-019-13818-7>. In the current manuscript they do the same for an eukaryotic model organism *S. cerevisiae*. The eukaryotic expression system is slightly more complex than the prokaryotic and is divided into subcellular compartments. As *S. cerevisiae* is a commonly used model organism, necessary data (e.g. Kcat values for enzymes) has been reasonably well available. The ETFL approach extends the simulations of metabolism to considering limitations arising from the catalytic capacities of enzymes and expression systems resulting in ability to predict some growth behaviours such as overflow metabolism without ad hoc constraints. ETFL approach also integrates metabolite and enzyme concentrations as variables facilitating omics data integration. Although the metabolites and enzyme concentrations are introduced as variables, the ETFL models do not describe the metabolic dynamics but allow simulations of metabolic steady states. ETFL approach is similar to previous ME-models that also integrate expression to the metabolic models. However, the ME-model do not integrate thermodynamic constraints, they have not yet been published for eukaryotic species, and they have a performance shortcoming compared to ETFL approach. ETFL approach discretizes growth for solving locally linearized MILP problems instead of NLP problems of ME-models.

The ETFL approach extended to a eukaryote is highly interesting for the field to be used as a tool for both applications and basic research. I have only a couple of minor concerns on the manuscript mainly on clarifications that I think would improve the appreciation of this manuscript independent of the previous publication on the ETFL model for *E. coli*. Please, find my detailed concerns below.

Rows 166-167: Please, clarify already here how the variable biomass composition was implemented.

Rows 177-179: Please, clarify how the ETFL approach allowed integration of the allocation constraint not compatible with ME-models.

Rows 213-214: Please, revise, a word is missing.

Authors response to the editor and reviewers' comments

Title: A genome-scale metabolic model of *Saccharomyces cerevisiae* that integrates expression constraints and reaction thermodynamics

Authors: Oftadeh et al.

Tracking #: NCOMMS-21-07186

We are thanking the reviewers and the editor for their very insightful, constructive, and positive comments. We have highlighted below in blue our detailed responses to the proposed improvements, suggestions, and modifications.

Additional changes: During the review process, we have continued the curation of the yeast genome-scale model, improving the modeling of the proton transfer through the electron transport chain. Therefore, in this review, in addition to our responses to the reviewers' comments, we have also re-run our computational simulation with the new curated stoichiometry of the genome-scale model. The details of the further curation are explained in the Supplementary Information (section: Thermodynamic curation of the *S. cerevisiae* genome-scale model Yeast8). The obtained results are marginally different than the ones presented in the previous version of the manuscript. However, as it can be seen in the reviewed manuscript, all the conclusions from the first version of the manuscript still hold. In the new results, the maximum growth of E(T)FL.cb and E(T)FL.vb models is 0.44 instead of 0.46 h⁻¹ and 0.41, instead of 0.42 h⁻¹, respectively. The change in the emergence of the Crabtree effect was also in the similar order, i.e., from 0.38-0.42 h⁻¹ to 0.35-0.39 h⁻¹ for different models. It is worth noting here that the new results are closer to the experimental values, highlighting that the new curations performed in the original Yeast8 model have improved the predictions of the yETFL models.

Reviewer #1 (Expertise: FBA, MILP, metabolic networks):

In this manuscript, Oftadeh et al present a metabolic-expression (ME) model for *Saccharomyces cerevisiae*. ME models are extensions of genome-scale metabolic (M) models, which in addition to metabolic reactions, account for gene expression processes related to metabolism and cellular growth. This work is an extension of a new ME model formulation referred to as Expression and Thermodynamic Flux (ETFL) for prokaryotes, which was recently published by the same group and applied to *E. coli*. In this study, the authors extend ETFL to eukaryotes and demonstrate its applicability by constructing a ME model for *S. cerevisiae*. This extension (named yETFL) accounts for additional ribosomes and RNA polymerases within mitochondria and for the fraction of proteins allocated to metabolism and growth. The authors verified the accuracy of this model by predicting the growth of *S. cerevisiae* for different glucose uptake rates and by performing gene essentiality analysis, and comparing with experimental data, and reported close agreements between the two. Additionally, the authors demonstrated that this model is able to capture the Crabtree effect without addition of any ad hoc constraints to the formulation. The novel contribution of this paper is proposing the first ME model for a eukaryotic cell, which paves the way for constructing such models for other eukaryotes as well. Additionally, given the wide applicability of *S. cerevisiae* for both basic and applied science and for industrial applications, the constructed ME model in this study will be of interest to a wide range of audience. **Overall, this is a well-written paper and a step in the right direction.** Addressing the following concerns can improve the quality of the manuscript:

1. Given that a whole-cell model of *S. cerevisiae* already exists (Ye et al, Biotechnol Bioeng,

2020, PMID: 32022245), at least a descriptive comparison of the yETFL model and this existing model seems necessary.

To address the reviewer's remark, we have added the following text in the Section "Expression and thermodynamics-enabled flux model of *S. cerevisiae*":

The recently developed whole-cell model of *S. cerevisiae*, WM_S288C, decomposes cell functionality into 26 cellular processes⁵⁶. yETFL includes three of those cellular processes, i.e., metabolism, RNA transcription, and protein translation model, which makes WM_S288C broader in scope. A comparison of the common parts between both models shows that both approaches use a constraint-based framework to model metabolism, but they differ in the form they model expression. While WM_S288C uses Ordinary Differential Equations (ODEs), yETFL uses a constraint-based optimization framework. The simplified approach used in yETFL allows for an efficient analysis of cellular behavior for different physiological conditions and different strains overcoming the requirement of the vast number of biophysical parameters present in WM_S288C and which are highly dependent on the strain and the environmental conditions⁵⁷. Furthermore, yETFL is able to simulate cellular processes under the macroscopic steady-state assumption and study the cell behavior in time intervals spanning a few hours³⁵, with a reasonable computational effort.

2. Given that (y)ETFL is a new ME model formulation, a head-to-head comparison of how different biological processes are formulated here compared to existing ME models (refs 11, 12 and 14 of the manuscript) could be very helpful. For example, it is not clear whether (y)ETFL accounts for reactions related to transcription and translation initiation or operon structures ("transcription units") implemented in existing ME models. Additionally, references for synthesis reactions or expression constraints are missing in the original ETFL paper (ref 13) making it unclear as to whether these were taken from existing ME models (refs 11, 12, 14) or were proposed for the first time in that paper. The authors can provide a summary of the processes or features in existing ME models that are not accounted for in (y)ETFL or vice versa. Additionally, for processes that are accounted for in both formulations, they can comment on how their treatment/formulation might differ between the two modeling paradigms.

The translational and transcriptional processes in yETFL are the same as in the original ETFL. Neither of the two models includes operon structure and transcription and translation initiation. Mechanistic detail about the "transcription units" can be included in the ETFL formulation by expanding around the corresponding existing transcription steps, as it was done in the ME model (ref 14). We are currently implementing these for the *E. coli* ETFL model. However, given that the corresponding processes for yeast and eukaryotic systems are more complex, we chose not to implement these in yETFL at this point. We have added a comment about this in the main text. These "transcription units" are currently "lumped" in existing parameters (such as transcription initiation) and this is a common modeling practice in gene expression models. Moreover, for the studies here and for most of the studies we can foresee using yETFL, such mechanistic details will not be necessary.

The transcription and translation reactions in ETFL formulation are derived from first principles based on mass balances, which are the same as in the ME models. However, in the ME models, the authors apply some reformulations which eliminate the explicit description of mass balances. Therefore, although all ME-models (refs 11-14 and the present work) have similar scope and are based on same first principles regarding the processes included in the model, the transcription and translation reactions presented in ref 13 and this work will correspond to different final equations. The details on these reactions are in the

Supplementary files of these studies (refs 11, 12, 13, and 14). And we have cited accordingly in the main text.

3. Can the proposed ME model capture the relative catalytic efficiency of isozymes? While in metabolic networks only the presence or absence of the enzymes are accounted for, we know that, in practice, the catalytic efficiency of the isozymes could be very different, i.e., the reaction rate could be different if one isozyme is removed compared to the other. How about when both isozymes are present? Will they have an additive effect on the reaction rate in (y)ETFL?

In ETFL, we account for the catalytic efficiency of each enzyme based on its turnover number (k_{cat}). A reaction might be catalyzed by different enzymes (i.e., isozymes) and as the reviewer suggested, the efficiency of these isozymes might be highly different. For example, if a reaction i is catalyzed by two isozymes j and j' , the corresponding catalytic constraint is:

$$v_i \leq k_{cat,j}E_j + k_{cat,j'}E_{j'}$$

where $k_{cat,j}$ and $k_{cat,j'}$ reflect different catalytic efficiency of the isozymes. This way, the maximum catalytic capacity of the reaction is defined as the sum of the maximum catalytic capacities of individual isozymes. A comment was added to the manuscript to clarify this (Line 347-355).

4. Line 74: ETFL does not “avoid” bilinear terms. It just approximates bilinear terms with a zeroth-order piecewise linear model.

Corrected according to the reviewer’s suggestion.

5. Equation (6): Each modeling metabolite i (e.g., carbohydrate) is produced in the pooling reaction and is consumed in the biomass reaction. These are taken care of by sign of the stoichiometric coefficients (positive in the former and negative in the latter). Therefore, the minus sign in this equation (before the second term) should be changed to a plus sign.

Corrected according to the reviewer’s suggestion.

6. Carbohydrates, lipids and ions (line 406): It is not clear how the growth-dependent biomass composition is implemented in the formulation. This sounds like an “if else” condition, e.g., if $0 \leq \mu \leq a$ then use these fractions else if $b < \mu \leq \mu_{max}$ use these other fractions, which requires defining new binary variables. Is this how these are implemented this in (y)ETFL?

We do not need additional binary variables for carbohydrates, lipids and ions. Indeed, one binary variable per each bin is sufficient to consider the growth dependence of all different components of the biomass, including DNA, RNA, protein, carbohydrate, lipid and ion. The growth dependence of carbohydrates, lipids, and ions is implemented in a similar way to the one of DNAs, RNAs, and proteins in the original ETFL formulation. Specifically, we first discretize the growth into bins and then we use a MILP optimization to determine the discretized growth value (bin) that corresponds to the studied physiology.

7. Is the mass fractions constraint (Equation 2) for E(T)FL_cb added as a constraint to the formulation?

Equation 2 does not exist as an explicit constraint in the formulation. Instead, as described in the manuscript, we introduced two explicit constraints (Eqs 3-4) that ensure that Equation 2 is satisfied. These equalities hold for both ETFL.cb and ETFL.vb at different growth rates. A comment was added to the manuscript accordingly.

8. Looking at the original ETFL formulation (ref 13), I see that the authors include total

capacity constraints for ribosome and RNA polymerase but I do not see any such constraints for metabolic enzymes (Enz_j). Is this a simple error or these constraints are not considered for metabolic enzymes for some reasons?

For ribosome and RNA polymerase, we have an explicit assignment between active enzymes and free enzymes. In the case of the metabolic enzymes, we do not have such assignment between complexes and free enzymes. Such description is unnecessary, as we do not account for enzyme mechanisms (e.g., Michaelis-Menten). However, the enzyme mass balances constrain directly only the total amount of the enzyme, as this is used in the flux inequality constraints based on the generalized maximum flux (i.e., $V_{max} = k_{cat}E_{total}$).

9. Table 3's caption: I think it is "Mathews" not "Mathew's" correlation coefficient. We thank the reviewer for identifying this typo.

Reviewer #2 (Expertise: Metabolic models, metabolic engineering):

In their manuscript, Oftadeh et al. have translated their ETFL modeling approach to *S. cerevisiae*: yETFL. This is a significant advance, as it is the first ME model of *S. cerevisiae* (although see my later comment). The major value of this model would be reproducible reuse by other research groups, who might want to address a wide variety of research questions, **but unfortunately for this purpose the model is inaccessible and significant detail on how the model was reconstructed is missing. In addition, the applications and validations shown are not particularly advanced, similar results have been shown with other formulations before and would rather be suited in Supplementary Material as necessary validation. The paper would improve significantly if the authors would be able to show one or several use cases where the model is providing new insight in yeast metabolism and its regulation.**

We provided more details on the construction of the models in the Supplementary Information. The code to construct the models is now available in a separate repository with more instructions on running the codes. As an example, a preconstructed yETFL model is provided in the repository (https://github.com/EPFL-LCSB/yetfl/tree/yetfl_model/output_model). However, we recommend the community to generate their own models with the code. This way, they can customize and adjust the model parameters such as k_{cat} s. The growth and overflow metabolism curves are used as quality control simulations that allow comparing our method's performance with similar methods.

The main aim of this manuscript is to present the yETFL model and the extension to the ETFL formulation that enables the construction of eukaryotic ME-models.

We are currently working on the metabolic engineering of yeast for the production of chemicals, and we are using yETFL in these studies. Preliminary results suggest that yETFL is useful to account for the cost and resource reallocation of overexpression of heterologous enzymes. These studies also use experimental data. However, we feel that including these studies here will be beyond the scope of this work.

1. Line 87: “no ME-model of *S. cerevisiae* has been developed”, while this is true to my knowledge, there is a whole-cell model WM_S288C (doi:10.1002/bit.27298) which one could include an ME-model as part of it. While the scope and application is different, it is worthwhile to mention and to discuss it in relation to yETFL.

Please see our response to comment 1 of Reviewer 1.

2. Line 104 (and 331): it is unclear what curations were applied to Yeast8. Thermodynamic properties were gathered and used in the TFA part, but what curations were done on the model itself? The GitHub repository contains `yeast8_thermo_curated.mat`, but the authors should provide a list of curations in supplementary material. A detailed explanation of the curation performed on the Yeast 8 was added to the Supplementary text, in the section Thermodynamic curation of the *S. cerevisiae* genome-scale model Yeast8.

3. Line 137: it is unclear why only two possible compositions of ribosomes were modelled, are there not many different possible combinations of subunits?

We agree with the reviewer and we thank them for pointing to the need for clarification. We have added the following text to the Ribosome section in Materials and Methods:

“The two modeled ribosomes represent only two out of many possible combinations of the peptides from sets A and B. Implementing these 2^{59} possibilities is currently

computationally intractable. Here, as a first approximation, we decided to keep the two ribosomal compositions because we wanted to be inclusive of all ribosomal genes while having a realistic production cost. The other possible combinations can be readily integrated into the model by adding similar constraints to what we have already included for the two modeled ribosomes. “

As a preliminary test on the sensitivity of the ribosomal composition on our findings, we randomly exchanged some of the genes in set A with their alternatives in set B, and the results were not impacted. It is also worth mentioning it is for the first time that multiple ribosomes are modeled in the ME models and therefore, we started with the simplest case. This could be further expanded to multiple combinations by adding sets of similar equations for each possible combination.

4. Line 148: why is rich growth medium used to define μ_{\max} in the model, as further model simulations are performed with minimal medium? Does rich medium not imply that many enzymes will not be required to sustain growth, and therefore give an underestimation of enzyme demand when simulated with minimal medium?

We use μ_{\max} to define the possible range of the growth rate which is then used to discretize the growth when we replace the bilinear terms in the formulation. We used μ_{\max} from the rich medium as a conservative upper bound on growth rate assuming that the growth on the rich medium is the highest growth rate that the organism can achieve in a normal condition. However, it is worth mentioning that this parameter can be decreased or increased by the users based on their needs to have a higher resolution or a higher range of the growth variation, respectively. A comment was added to the manuscript to clarify this (Line 157-163).

5. Line 175: please clarify that “previously reported ME-models” refers to ME-models in general, not a non-existent *S. cerevisiae* ME-model.

Clarified as suggested by the reviewer.

6. Line 186: please document in Supplementary Material what parameter values were used. This also refers to any other parameter that was set in yETFL, the Materials and Methods section does not give sufficient information on this. This was added to the Supplementary file (Table S1-6).

7. Line 194: verb missing.

Corrected as suggested by the reviewer.

8. Line 207: unclear what is meant with “However, this is not comprehensive of *S. cerevisiae* genes”

This sentence was modified as follows:

“However, these 5061 genes do not include all *S. cerevisiae* genes.”

9. Line 226: swap to “looked next”

It was edited as suggested by the reviewer.

10. Line 265: in the introduction GECKO is mentioned as the closest alternative to yETFL. What is missing from the discussion is then how yETFL is better than GECKO? The results in the manuscript show that yETFL performs as well as GECKO (albeit without saturation factor), but what are things yETFL can do / predict / analyze that cannot be done by the GECKO model?

Gecko is a coarse-grained framework with phenomenological constraints for proteome limitation. In contrast, ETFL is a fine-grained framework that accounts for proteome limitations mechanistically by integrating additional processes, such as transcription and translation. As a result, ETFL can predict parameters such as growth-condition dependent biomass composition as well as transcription and translation machinery content, which cannot be done by GECKO. Moreover, the mechanistic representation of the expression system provides an additional possibility to ETFL to simulate the perturbations on the expression machinery, RNA transcripts, and gene copy numbers. Finally, in addition to proteomics data, which can also be integrated into GECKO, ETFL enables the integration of transcriptomics data. These were added to the manuscript accordingly.

11. Line 293: the Materials and Methods section is not detailing all relevant information. It is probably most suitable to provide this in Supplementary Material instead. This includes what curations were performed on Yeast8 and what parameters were set, as both mentioned above. The data provided on Zenodo seems like a random selection from the whole GitHub repository (why not have the whole repository backed-up on Zenodo?). Why is the curated model distributed as MATLAB file, particularly considering pyTFA runs on python? The thermo_data_yeast8.mat file does not have sufficient meta data explaining its content. There should be a much more detailed explanation of data presentation and how this was used in establishing yETFL.

The thermodynamic curation is now described in more detail in the Supplementary Information, in the section Thermodynamic curation of the *S. cerevisiae* genome-scale model Yeast8. The extension to the ETFL formulation was described in Supplementary file, the section Implementing multiple expression systems. The full list of constraints and parameters used in yETFL was provided in Tables S1-S7. Also, the data was moved completely to Zenodo and GitHub only contains the code. The explanation was added to describe the content in Zenodo.

12. Line 526: having a separate branch, with even dev_ in the name, does not seem like a good solution.

As suggested by the reviewer, we moved the code for creating the yETFL models to a separate repository (<https://github.com/EPFL-LCSB/yetfl>). The changes in the main ETFL repository were merged to the master branch (<https://github.com/EPFL-LCSB/etfl>). This is also stated in the text.

13. GitHub: I was unable to run the code (Ubuntu 20.04 on WLS2, python 3.8.5). pipenv freeze output is included at the end of the review. When following the instructions in README_yETFL.txt, when subsequently running tutorials/helper_gen_models_yeast.py gives the following error message:

```
Traceback (most recent call last):
File "helper_gen_models_yeast.py", line 16, in <module>
from etfl.core import Enzyme, Ribosome, RNAPolymerase, ThermoMEModel, MEModel
File "/home/XXXX/yETFL/etfl-dev_yetfl/etfl/core/__init__.py", line 1, in <module>
from .enzyme import Enzyme, Ribosome, RNAPolymerase
File "/home/XXXX/yETFL/etfl-dev_yetfl/etfl/core/enzyme.py", line 19, in <module>
from Bio.Alphabet import DNAAlphabet, ProteinAlphabet
File "/home/XXXX/.local/share/virtualenvs/yETFL-gNQIO2nY/lib/python3.8/site-
packages/Bio/Alphabet/__init__.py", line 20, in <module>
```

raise ImportError(
ImportError: Bio.Alphabet has been removed from Biopython. In many cases, the alphabet
can simply be ignored and removed from scripts. In a few cases, you may need to specify the
``molecule_type`` as an annotation on a SeqRecord for your script to work correctly. Please
see <https://biopython.org/wiki/Alphabet> for more information.

Alternatively, a Docker container is provided, although wide-spread use of (y)ETFL is not
helped if it only runs as Docker. Regardless, following the instructions (on the master branch
of the repository) result in an error message when “. run” is run in etfl/docker:

```
ERROR: File "setup.py" not found. Directory cannot be installed in editable mode:
/src/optlang
```

Therefore, I have not been able to run the code and have therefore not been able to evaluate
its reproducibility.

```
##### pip freeze output:
appdirs==1.4.4
attrs==20.3.0
biopython==1.78
bokeh==2.3.0
certifi==2020.12.5
cobra==0.21.0
colorama==0.4.4
commonmark==0.9.1
decorator==4.4.2
depinfo==1.7.0
diskcache==5.2.1
# Editable install with no version control (ETFL==0.0.2)
-e /home/XXXXX/yETFL/etfl-dev_yetfl
future==0.18.2
h11==0.12.0
httpcore==0.12.3
httpx==0.17.1
idna==3.1
importlib-resources==5.1.2
iniconfig==1.1.1
Jinja2==2.11.3
MarkupSafe==1.1.1
mpmath==1.2.1
networkx==2.5
numpy==1.20.1
optlang==1.4.4
packaging==20.9
pandas==1.2.3
Pillow==8.1.2
pluggy==0.13.1
py==1.10.0
```

pydantic==1.8.1
Pygments==2.8.1
pyparsing==2.4.7
pytest==6.2.2
pytfa==0.9.3
python-dateutil==2.8.1
python-libsmb==5.19.0
pytz==2021.1
PyYAML==5.4.1
rfc3986==1.4.0
rich==6.2.0
ruamel.yaml==0.16.13
ruamel.yaml.clib==0.2.2
scipy==1.6.1
six==1.15.0
sniffio==1.2.0
swiglpk==5.0.3
sympy==1.7.1
toml==0.10.2
tornado==6.1
tqdm==4.59.0
typing-extensions==3.7.4.3
xlrd==2.0.1

We would like to thank the reviewer for making us aware of this problem. The error is related to one of the ETFL's dependencies, Biopython. This package has recently published a new version (1.78) in which the module Bio.Alphabet is entirely removed. This has caused troubles for different biological packages in python that relies on this module. We updated the ETFL code to remove dependency on this submodule. Since we also moved the yETFL code from the main repository to a separate repository, we suggest that you uninstall the ETFL package, fetch the updated repositories (both ETFL and yETFL) and reinstall it. In this way, the problem should be solved.

Also, the problem with the Docker file was resolved and the container can be created successfully.

Reviewer #3 (Expertise: yeast genome scale metabolic models):

The Authors have previously presented an approach to extend genome-scale metabolic models with expression and thermodynamic constraints for a prokaryotic model organism *E. coli* <https://www.nature.com/articles/s41467-019-13818-7>. In the current manuscript they do the same for an eukaryotic model organism *S. cerevisiae*. The eukaryotic expression system is slightly more complex than the prokaryotic and is divided into subcellular compartments. As *S. cerevisiae* is a commonly used model organism, necessary data (e.g. K_{cat} values for enzymes) has been reasonably well available. The ETFL approach extends the simulations of metabolism to considering limitations arising from the catalytic capacities of enzymes and expression systems resulting in ability to predict some growth behaviours such as overflow metabolism without ad hoc constraints. ETFL approach also integrates metabolite and enzyme concentrations as variables facilitating omics data integration. Although the metabolites and enzyme concentrations are introduced as variables, the ETFL models do not describe the metabolic dynamics but allow simulations of metabolic steady states. ETFL approach is similar to previous ME-models that also integrate expression to the metabolic models. However, the ME-model do not integrate thermodynamic constraints, they have not yet been published for eukaryotic species, and they have a performance shortcoming compared to ETFL approach. ETFL approach discretizes growth for solving locally linearized MILP problems instead of NLP problems of ME-models. **The ETFL approach extended to a eukaryote is highly interesting for the field to be used as a tool for both applications and basic research.** I have only a couple of minor concerns on the manuscript mainly on clarifications that I think would improve the appreciation of this manuscript independent of the previous publication on the ETFL model for *E. coli*. Please, find my detailed concerns below.

Rows 166-167: Please, clarify already here how the variable biomass composition was implemented.

As suggested by the reviewer, the following sentence is added in the manuscript:

“To account for this variation, the fractions and hence, stoichiometric coefficients of each biomass building block are determined based on experimental data. This way, we obtain a set of different biomass reactions, each associated with a specific growth rate. Then, we use a MILP optimization problem to determine which of the biomass reactions corresponds to the studied physiology.”

Rows 177-179: Please, clarify how the ETFL approach allowed integration of the allocation constraint not compatible with ME-models.

We agree with the reviewer that such clarification is needed. After reconsidering, we removed that statement because we realized that this allocation constraint can also be implemented in the previous formulation of ME-models, but it will require a reformulation as the ME models apply some modifications/reformulations in the mass balances to improve the computational efficiency. In ETFL, such reformulations are unnecessary as the ETFL formulation allows for computational efficiency.

Rows 213-214: Please, revise, a word is missing.
Corrected according to the reviewer's comment.

Reviewers' Comments:

Reviewer #1:

Remarks to the Author:

The authors have done a good job addressing concerns raised about the original manuscript, however, there are a few places, which require further attention by the authors:

1. Authors response to my original concern 2 about comparison of (y)ETFL formulation with existing ME model formulations has clarified differences between these two modeling paradigms to some extent but this response does not seem to be included anywhere in the revised manuscript.

Additionally, just citing papers on previous ME model formulations is not enough for a full comparison of the two modeling frameworks. The authors need to

- a. briefly discuss these differences in the main text of the revised manuscript,
- b. provide a more detailed comparison in the supplementary text. For example, statements like "However, in the ME models, the authors apply some reformulations which eliminate the explicit description of mass balances." or "..., the transcription and translation reactions presented in ref 13 and this work will correspond to different final equations" are rather vague and need to be elaborated in the supplementary text, e.g., by explaining "reformulations" and by comparing example reaction equations that are different between the two modeling frameworks. Furthermore, as noted in my original concern 2, processes that might have been accounted for in previous ME models (such as translation/transcription initiation) but not in (y)ETFL explicitly or vice versa (e.g., thermodynamic constraints) should be fully listed.

I feel that a full comparison is necessary to establish (y)ETFL as an alternative formulation for ME models in the systems biology community.

2. Authors' response to my original concern (8) regarding active and free metabolic enzymes should be added somewhere to the revised manuscript as there is no way for readers to guess why the total capacity constraints are written for ribosome and RNA polymerase but not for metabolic enzymes.

Reviewer #2:

Remarks to the Author:

The author's response was satisfactory.

Reviewer #3:

Remarks to the Author:

My concerns have been adequately addressed in the revised version of the manuscript, and I recommend publishing it.

Authors response to the editor and reviewers' comments

Title: A genome-scale metabolic model of *Saccharomyces cerevisiae* that integrates expression constraints and reaction thermodynamics

Authors: Oftadeh et al.

Tracking #: NCOMMS-21-07186A

We would like to thank the reviewers again for their comments and feedbacks which helped us to improve our work. We have highlighted below in blue our detailed responses to the remaining concerns.

Reviewer #1 (Remarks to the Author):

The authors have done a good job addressing concerns raised about the original manuscript, however, there are a few places, which require further attention by the authors:

1. Authors response to my original concern 2 about comparison of (y)ETFL formulation with existing ME model formulations has clarified differences between these two modeling paradigms to some extent but this response does not seem to be included anywhere in the revised manuscript. Additionally, just citing papers on previous ME model formulations is not enough for a full comparison of the two modeling frameworks. The authors need to

- briefly discuss these differences in the main text of the revised manuscript,
- provide a more detailed comparison in the supplementary text. For example, statements like “However, in the ME models, the authors apply some reformulations which eliminate the explicit description of mass balances.” or “..., the transcription and translation reactions presented in ref 13 and this work will correspond to different final equations” are rather vague and need to be elaborated in the supplementary text, e.g., by explaining “reformulations” and by comparing example reaction equations that are different between the two modeling frameworks. Furthermore, as noted in my original concern 2, processes that might have been accounted for in previous ME models (such as translation/transcription initiation) but not in (y)ETFL explicitly or vice versa (e.g., thermodynamic constraints) should be fully listed.

I feel that a full comparison is necessary to establish (y)ETFL as an alternative formulation for ME models in the systems biology community.

The following was added to the main text (Line 371-386):

“ETFL¹³ is different from the other formulations of ME-models^{12,14} in several aspects. On the one hand, only ETFL allows the integration of thermodynamic constraints and metabolomics data. Also, multiple expression systems were implemented only in ETFL. On the other hand, stable RNA splicing and operon structure were considered in the previous formulations of ME-models, but not in ETFL (see Supplementary Notes for further details). The inclusion of ionic cofactors to form functional enzymes, i.e., metalloproteins, is partially considered in the previous formulations of ME-models, whereas ETFL lumps such requirements in the biomass reaction, alike the FBA models. Both formulations derived the mass balances for the macromolecules from the first principles, but rather in a different way. For example, in the other formulations of ME-models the enzymes are involved in the reactions as metabolites. In ETFL, however, the enzymes are coupled to the reactions based on their catalytic capacity (see Supplementary Notes and Supplementary Table S3).

Finally, ETFL is a MILP formulation which can be solved by conventional double-precision solvers, while the previous formulations of ME-models were nonlinear and only solvable with special quad-precision solvers.”

In addition, the following was added to the Supplementary Notes:

“In this section, we provide a general comparison between the ETFL²⁵ and the other ME-model formulations^{23,24}.

A key difference between the two formulations is that ETFL enables the integration of thermodynamic constraints and metabolomics data. Moreover, ETFL can account for multiple expression systems. On the other hand, the stable RNA splicing, and transcription initiation are only formulated in the other ME-model formulations. These functions can be also included in ETFL by expanding the formulation around the related processes. The transcription initiation is not completely neglected in ETFL but instead lumped in the other processes. This lumping is similar to the lumping used for translation initiation and elongation in all ME-models. Since it was shown that including such processes does not impact the predictions of the model in the protein limitation studies²⁴, and the transcription initiation is more complex in eukaryotic organisms, we did not model them at this stage. Also, in the other formulations of ME-models, the formation of metalloproteins is explicitly considered. In ETFL, the ionic requirements are lumped in the biomass reaction as it was done in the FBA models.

There were two formulations in the other ME-models regarding the enzyme mass balances. In the first formulation, presented in O'brien *et al.*²³, the enzymes appeared as metabolites in the reactions, and to properly account for mass balances, two pseudometabolites per enzyme were added to the model. Consider this toy example, where the following reaction with flux v_1 is catalyzed by enzyme E :

To associate the enzyme abundance to the flux v_1 , the following reactions were added:

where coupling and $\text{enzyme}_{\text{prime}}$ are pseudometabolites introduced to facilitate the addition of constraints and $\alpha = \frac{\mu}{k_{\text{cat}}}$. While the mass balance constraint associated with $\text{enzyme}_{\text{prime}}$ is an equality, the mass balance for coupling is an inequality. Writing the mass balances for the enzyme and the pseudometabolites, the following constraints can be derived:

$$v_{\text{tnsl}} - v_{\text{deg}} - v_2 = 0 \quad (10)$$

$$v_1 - v_0 = 0 \quad (11)$$

$$v_2 \geq \alpha v_0 \quad (12)$$

where v_{tnsl} and v_{deg} are respectively the rates of translation and degradation of the enzyme.

In the second formulation of ME-models, presented in Lloyd *et al.*²⁴, to simplify the solving process, the inequality constraint in Equation 12 was converted to an equality. This assumes that all the enzymes catalyze their reactions with the maximum possible rate at the optimal solution. This way, the two pseudometabolites and their corresponding reactions were removed.

In the ETFL formulation, no pseudometabolite is considered and the enzymes are not involved in the reactions as metabolites²⁵. For the reaction presented by Equation 6, the following two constraints are imposed:

$$v_{\text{tnsl}} - v_{\text{deg}} - v_{\text{dil}} = v_{\text{tnsl}} - v_{\text{deg}} - \mu E = 0 \quad (13)$$

$$v_1 \leq k_{\text{cat}} E \quad (14)$$

where E is the total concentration of the enzyme and Equation 14 applies the generalized maximum flux (i.e., $V_{\text{max}} = k_{\text{cat}} E$). Although the constraints in Equations 13-14 differ from the ones in Equations 10-12 and the number of variables and constraints are different, it is straightforward to demonstrate that ETFL formulation and the formulation presented in O'Brien *et al.* are mathematically equivalent. That is, from Equation 10 and Equation 13, it follows that $v_2 = \mu E$. Then, by replacing v_2 in Equation 12, we obtain Equation 14.

The equality assumption in Equation 12 presented in Lloyd *et al.* would be equivalently modeled in ETFL by converting the inequality in Equation 14 to an equality. This implies that at the optimal solution, all the enzymes perform under complete saturation, which might be an unrealistic assumption in some cases.”

2. Authors’ response to my original concern (8) regarding active and free metabolic enzymes should be added somewhere to the revised manuscript as there is no way for readers to guess why the total capacity constraints are written for ribosome and RNA polymerase but not for metabolic enzymes.

Based on the reviewer’s suggestion, we added the following to the manuscript (Line 356-363):

“For the ribosomes and RNA polymerases, we model both free enzymes and enzyme complexes. For the metabolic enzymes, we do not account for enzyme mechanisms (e.g., Michaelis-Menten), and therefore making the distinction between free enzymes and enzyme complexes is not necessary. The enzyme mass balances constrain directly only the total amount of the enzyme, E_{total} . This quantity is used in the flux inequality constraints to express the maximal reaction velocity, $V_{\text{max}} = k_{\text{cat}} E_{\text{total}}$.”

Reviewer #2 (Remarks to the Author):

The author's response was satisfactory.

Reviewer #3 (Remarks to the Author):

My concerns have been adequately addressed in the revised version of the manuscript, and I recommend publishing it.